# Factors associated with weighing a child at birth: Evidence from 16 sub-Saharan African countries

Alex Bawuah[1], Samuel Ampaw[2], Edward Nketiah-Amponsah[3]*

1 Health Economics Research Unit, Aberystwyth Business School, Aberystwyth University, Aberystwyth, Ceredigion, Wales, United Kingdom, 2 Global Poverty Research Lab, Kellogg School of Management, Northwestern University, Evanston, Illinois, United States of America, 3 Department of Economics, University of Ghana, Legon, Ghana

* enamponsah@ug.edu.gh

## Abstract

### Background

Several children from sub-Saharan Africa (SSA) are not weighed at birth. The lack of birthweight data is a significant challenge in monitoring the global prevalence of extreme birthweight, either low or high, and newborn health. This data guides resource allocation and the design of targeted health policies to address neonatal complications and mortalities. This paper explores the demand-side predictors of newborn weighing.

### Methods

Data were obtained from the Demographic and Health Surveys (DHS) of 16 countries in SSA, conducted from 2014 to 2021. Multivariate logistic regression was used to achieve the study's objectives.

### Results

Approximately 59% of the study population were weighed at birth. This prevalence rate varied widely across the 16 countries, ranging from 23% in Chad to 94% in Gabon. The study documents a positive association between higher socioeconomic status and the probability of being weighed at birth. Specifically, older women and women with higher education and wealth were more likely to weigh their newborns at birth. Also, women who delivered at healthcare facilities and those who used antenatal care had a higher likelihood of weighing their children at birth. Urban residents were more likely to weigh their children at birth. On the contrary, the likelihood of weighing a child at birth decreases with parity.

**Data availability statement:** The data is owned by a third party – Demographic and Health Survey (DHS). The data is publicly available on the DHS website (www.dhsprogram.com). The data specific to our empirical analysis in this paper has been shared as supporting information

**Funding:** The author(s) received no specific funding for this work.

**Competing interests:** I have read the journal's policy and the authors of this manuscript have the following competing interests: no competing interest.

**Abbreviations:** OR, Odds Ratio; AOR, Adjusted Odds Ratio; CI, Confidence Interval; ANC, Antenatal Care; SSA, Sub-Sahara Africa; DHS, Demographic and Health Survey; WHO, World Health Organisation; IQ, Intelligence Quotient; VIF, Variance Inflation Factor; TI, Tolerance Indices

## Conclusion

The study highlights the need to target pregnant women of lower socioeconomic status for interventions aimed at averting severe morbidity and mortality occasioned by conditions of low birthweight.

## Background

Birthweight is the first weight of the foetus or newborn measured after birth [1]. It can be categorized into three: low birthweight (birthweight ≤ 2.5 kg), normal birthweight (birthweight between 2.5 kg – 4.0 kg), and macrosomia or high birthweight (birthweight > 4.0 kg) [2,3]. A child's birthweight is an essential indicator of their health and development, as it can affect their short- and long-term outcomes [4]. For instance, low birthweight babies are at a higher risk of developing chronic diseases [5], stunted growth [6], lower IQ [7], and reduced life expectancy [8]. High birthweight is associated with high risks for certain malignancies in childhood, hypertension in childhood, psychiatric disorders, type 1 and 2 diabetes, breast cancer, and obesity [9–11].

Given the risk factors associated with high and low birthweight, the WHO recommends that babies be weighed during postnatal care [12]. Weighing children at birth has several merits. First, knowledge about the birthweight provides vital information about a child's health [5]. It guides medical care and interventions, such as medications and feeding plans, to promote the best possible outcomes for the child's health and development. For instance, babies with low or high birthweight may need to be fed differently or be given particular medications compared to babies with normal birthweight. Second, weighing newborns helps monitor the prevalence of low or high birthweight [8]. Third, a child's birthweight can give important information about the mother's health. Studies have shown that mothers who have poor nutrition, anaemic conditions, and are underweight are more likely to have low birthweight babies [13,14]. Fourth, studies relating to a child's birthweight have often been centred around the factors that influence a child's birthweight [15–20]. Researchers were able to conduct these studies because the children were weighed at birth and their birthweights recorded. These studies could not have been undertaken if the children had not been weighed at birth.

The challenge, however, is that many newborns are not weighed at birth. For instance, in 2015, nearly one-third of newborns (39.7 million) globally were not weighed at birth, with Africa recording the highest percentage of newborns without a recorded birthweight (51.7%, or 21.5 million) [8]. This situation places a significant challenge on monitoring the prevalence of low or high birthweight and the health of newborns due to the lack of birthweight data for many children. This study, therefore, aims to investigate the factors associated with weighing a child at birth in sub-Saharan Africa (SSA).

Various factors, including individual sociodemographic, maternal, household, and community-level factors, can influence the decision to weigh a child at birth. Understanding these factors can guide policymakers in implementing policies that will

encourage the weighing of newborns. To the best of our knowledge, there has been no published literature on this topic. This study is the first to explore the factors influencing the decision to weigh newborns.

## Methods

### Data source

The study used data from the Demographic and Health Surveys (DHS) of 16 countries in SSA conducted from 2014 to 2021 (these are countries with high neonatal mortality rates for which data were available). The DHS is a nationwide survey conducted in over 85 low- and middle-income countries worldwide and follows a consistent protocol and terminology across all countries [21]. It employs a structured questionnaire to gather information on various health indicators, including maternal and child health, fertility, family planning utilization, morbidity, and mortality [21]. The DHS uses a two-stage sampling technique to collect data, starting with selecting enumeration areas based on each country's sampling frame, followed by the selection of households from each enumeration area. Detailed information on the sampling and data collection methods can be found in the work of Aliaga and Ren [22].

This study employed the children's dataset (KR file) from the DHS surveys (see S1 Appendix). The KR file contains a single entry for each child born to the interviewed women within the five years preceding the survey [23]. The total sample size of children (aged 0–5 years) from the sixteen countries in the specified duration was 183,541.

### Variables

The study's outcome variable is whether the child was weighed at birth. In the survey, the respondents were asked if the child was weighed at birth. The response was coded as "1 = Yes" and "0 = No" for children who were weighed and those who were not weighed at birth, respectively.

The study used a total of 13 explanatory variables. The variables include the respondent's age, woman's highest level of education, marital status, residence, parity, decisions about woman's health, used antenatal care (ANC), place of delivery, covered by insurance, household wealth status, birth type, sex of the child, currently working status and distance to a healthcare facility. Respondent's age was coded on a five-interval scale from 15 to 49. Women's education was coded as "none", "primary", "secondary", and "higher". Marital status was coded as "married" and "otherwise". The residence was coded as "rural" and "urban". Parity indicates the number of times the woman has had a live birth. Decisions about women's health were coded as "woman alone", "together with partner/husband", and "husband/partner alone". ANC use was coded as "yes" and "no". The place of delivery (childbirth) was coded as "health facility" and "otherwise". Covered by insurance was coded as "yes" and "no". Household wealth status was coded as "poorest", "poorer", "middle", "richer", and "richest". The birth type was coded as "single birth" and "multiple births". Child sex was coded as "male" and "female". Distance to healthcare facilities was coded as "a big problem" and "not a big problem" (in the survey, the women were asked whether distance to the healthcare facility was a big problem or not a big problem).

The year variable (the year in which the survey was conducted) was initially included in the model to account for potential temporal effects. However, we found that it was perfectly collinear with the country variable. Given this collinearity, the year variable was omitted from the final model. Furthermore, the model did not include birth order due to collinearity with the parity variable.

### Data analysis

The data were analysed using STATA version 16. Frequency tables were used to describe the data (Table 2). Given the binary nature of the dependent variable, the study applied logistic regressions to evaluate the determinants of measuring a child's weight at birth [24]. First, a bivariate logistic regression analysis was conducted to estimate the unadjusted odds ratios (ORs) for each determinant. Afterwards, a multivariable logistic regression was performed to estimate the adjusted odds ratios (AORs) for all determinants, considering clustering (at the country level) and

sampling weights. We handled missing data using listwise deletion; thus, observations with missing values in any of the variables included in the regression model were excluded from the analysis. This approach ensures consistency across all model estimates.

Prior to the regression analysis, we conducted a diagnostic test for collinearity between the independent variables by looking at the variance inflation factor (VIF) and tolerance indices (TI) of the independent variables. VIF and TI are measures used to assess the level of multicollinearity between the $i$th independent variable and other independent variables in a regression model [25]. VIFs above 5 and TIs less than 0.20 are considered signs of multi-collinearity [26]. The VIFs for all the independent variables used in the study were below 5. Similarly, the TIs were above 0.20. These results are provided in S2 Appendix.

### Ethical consideration

The data used for this study, Demographic and Health Survey (DHS) is secondary, and its usage does not require ethical clearance. Measure DHS, which is responsible for undertaking standardized Demographic and Health Surveys in over 90 countries, duly obtained ethical clearance from the requisite institutions in all 16 countries included in this study, as well as the Institutional Review Board of the ICF International. In this regard, ethical approval and consent of participants to participate in the study are not applicable. It is worth noting that all data are anonymous and confidential, so there is no way of identifying individuals in the database.

## Results

### Descriptive statistics

As shown in Table 1, the sample size for the study was 183,541. Almost 60% of the children were weighed at birth. Rwanda had the highest percentage of children weighed at birth (95.12%), whereas Ethiopia had the lowest (27.48%). Most of the countries recorded a less than 70% prevalence of babies weighed at birth (Angola – 55.14%; Cameroon

**Table 1. Sample size characteristics.**

| Country | Year of survey | Number ofchildren | % of Children weighed at birth |
|---|---|---|---|
| Angola | 2015-2016 | 14322 | 55.14 |
| Benin | 2017-2018 | 13589 | 76.48 |
| Cameroon | 2018 | 9733 | 69.19 |
| Ethiopia | 2016 | 10641 | 27.48 |
| Gambia | 2019-2020 | 5447 | 63.81 |
| Ghana | 2014 | 8362 | 86.04 |
| Guinea | 2018 | 7951 | 51.06 |
| Kenya | 2014 | 9967 | 59.51 |
| Liberia | 2019-2020 | 5704 | 65.83 |
| Madagascar | 2021 | 12499 | 38.53 |
| Malawi | 2015-2016 | 9940 | 52.66 |
| Mali | 2018 | 17286 | 91.7 |
| Nigeria | 2018 | 33924 | 29.28 |
| Rwanda | 2019-2020 | 8092 | 95.12 |
| Senegal | 2019 | 6125 | 76.91 |
| Zambia | 2018 | 9959 | 83.2 |
| Total | | 183541 | 59.33 |

– 69.19%; Ethiopia – 27.48%; Gambia 63.81%; Guinea – 51.06%; Kenya – 59.51%; Liberia – 65.83%; Madagascar – 38.53%; Malawi – 52.66%; Nigeria – 29.28%).

Table 2 presents the results of the descriptive summary of the sample. It shows that most of the children are males (50.73%). Also, a higher proportion of the respondents are between 25 and 29 years old (26.79%). Likewise, most of the respondents were married (74.62%), rural residents (67.99%), had a single birth (96.47%), used ANC (87.16%), and reported that distance to a healthcare facility was not a big problem (61.02%). Most of the respondents did not have insurance (91.5%).

Furthermore, most of the children who were weighed at birth were delivered in a healthcare facility (91.71%), whereas those who were not weighed were delivered elsewhere (83.18%), mostly at home (81.74%). Similarly, a higher proportion of those who weighed their children had primary education (36.61%), whereas those who did not weigh their children had no education (59.18%). Likewise, a higher proportion of those who weighed their children belonged to the wealthiest wealth category (21.6%), whereas those who did not weigh their children belonged to the poorest category (39.95%). Also, the mean parity of those who did not weigh their children was higher (4.45) than those who weighed their children (3.53).

## Determinants of weighing children at birth

Table 3 presents the findings from both the bivariate and multivariable logistic regression analyses. Results of the multivariable logistic regression show that age, parity, place of delivery, residence, education, wealth status, the person(s) who decides on the woman's healthcare, ANC utilisation, distance to a health facility, and having insurance were significant determinants of weighing children at birth. The odds of weighing children at birth for respondents aged between 20–24, 25–29, 30–34, 35–39, 40–44, and 45–49 were respectively 1.14 (95% CI = 1.01–1.29), 1.30 (95% CI = 1.03–1.63), 1.57 (95% CI = 1.17–2.09), 1.85 (95% CI = 1.35–2.54), 1.91 (95% CI = 1.32–2.76), and 1.67 (95% CI = 1.14–2.44) times higher than those between the ages of 15–19. Furthermore, relative to the uneducated, the odds of weighing children at birth were 1.38 (95% CI = 1.19–1.60), 1.64 (95% CI = 1.41–1.91), and 2.68 (95% CI = 2.33–3.08) times higher for women with primary, secondary, and higher education, respectively. Similarly, the probability of weighing children at birth was 1.20 (95% CI = 1.07–1.35), 1.48 (95% CI = 1.22–1.78), 2.05 (95% CI = 1.56–2.70), and 3.91 (95% CI = 2.96–5.17) times higher for those from poorer, middle, wealthier and wealthiest households, respectively, compared to those from the poorest households. Women who delivered at healthcare facilities were more likely to weigh their children at birth (AOR = 52.29, 95% CI = 38.43–71.16). Similarly, women who used ANC during pregnancy were more likely to weigh their children at birth (AOR = 4.62, 95% CI = 3.76–5.67). Urban residents were more likely to weigh their children at birth (AOR = 2.29, 95% CI = 1.79–2.91) than rural residents. Also, the insured are more likely to weigh their children at birth (AOR = 1.32, 95% CI = 1.09–1.61). In addition, women who make decisions about their healthcare alone and those who decide jointly with their husbands/partners are 1.28 (95% CI = 1.04–1.57) and 1.22 (95% CI = 1.01–1.46) times more likely than women whose husbands/partners alone make decisions about their healthcare. Furthermore, those who reported that distance to the healthcare facility is not a big problem were more likely to weigh their children (AOR = 1.09, 95% CI = 1.00–1.19) than their counterparts who had difficulty accessing the healthcare facility. Moreover, the likelihood of weighing children at birth decreases as parity increases (AOR = 0.94, 95% CI = 0.90–0.98). Furthermore, children from other countries are less likely to be weighed compared to children from Rwanda.

The interactions between residence and place of birth showed that urban residents who give birth in healthcare facilities are 54.84 times more likely to weigh their children compared to rural residents who give birth at other places. Similarly, women who used ANC services and gave birth in a healthcare facility were 166.69 times more likely to weigh their children than those who did not use ANC and gave birth elsewhere. Also, compared to rural residents with no education, urban residents with secondary levels of education and urban residents with higher levels of education are 5.12 and 8.80 (respectively) times more likely to weigh their children.

**Table 2. Descriptive statistics of the sample.**

| Variable | Pooled Sample | Weighed | Not Weighed | |
|---|---|---|---|---|
| | Frequency (%) | Frequency (%) | Frequency (%) | $\chi^2$/ Diff.t-test |
| **Child's Characteristics** | | | | |
| Weighed at birth (*n* = 183541) | | | | |
| Yes | 108892 (59.33) | | | |
| No | 74649 (40.67) | | | |
| Missing = 0 (0.00%) | | | | |
| Child's sex (*n* = 183541) | | | | |
| Male | 93104 (50.73) | 55527 (50.99) | 37577 (50.34) | 0.006 |
| Female | 90437 (49.27) | 53365 (49.01) | 37072 (49.66) | |
| Missing = 0 (0.00%) | | | | |
| Birth type (*n* = 183541) | | | | |
| Single birth | 177062 (96.47) | 104721 (96.17) | 72350 (96.92) | <0.001 |
| Multiple births | 6473 (3.53) | 4173 (3.83) | 2301 (3.08) | |
| Missing = 0 (0.00%) | | | | |
| **Mother's Characteristics** | | | | |
| Age (*n* = 183541) | | | | |
| 15–19 | 11510 (6.27) | 6733 (6.18) | 4777 (6.40) | <0.001 |
| 20–24 | 41277 (22.49) | 24796 (22.77) | 16481 (22.08) | |
| 25–29 | 49173 (26.79) | 29188 (26.80) | 19985 (26.77) | |
| 30–34 | 38206 (20.82) | 23069 (21.19) | 15137 (20.28) | |
| 35–39 | 27072 (14.75) | 16233 (14.91) | 10839 (14.52) | |
| 40–44 | 12267 (6.68) | 6885 (6.32) | 5382 (7.21) | |
| 45–49 | 4036 (2.20) | 1988 (1.83) | 2048 (2.74) | |
| Missing = 0 (0.00%) | | | | |
| Level of education (*n* = 183539) | | | | |
| None | 73968 (40.30) | 29790 (27.36) | 44178 (59.18) | <0.001 |
| Primary | 59385 (32.36) | 39865 (36.61) | 19520 (26.15) | |
| Secondary | 43334 (23.61) | 33059 (30.36) | 10277 (13.77) | |
| Higher | 6853 (3.73) | 6179 (5.68) | 674 (0.90) | |
| Missing = 2 (0.00%) | | | | |
| Marital Status (*n* = 183541) | | | | |
| Married | 136961 (74.62) | 78307 (71.91) | 58654 (78.57) | <0.001 |
| Otherwise | 46580 (25.38) | 30585 (28.09) | 15995 (21.43) | |
| Missing = 0 (0.00%) | | | | |
| Residence (*n* = 183541) | | | | |
| Urban | 58753 (32.01) | 45620 (41.89) | 13123 (17.58) | <0.001 |
| Rural | 124798 (67.99) | 63272 (58.11) | 61526 (82.42) | |
| Missing = 0 (0.00%) | | | | |
| Parity (*n* = 183541) | *Mean* = 3.91 | *Mean* = 3.53 | *Mean* = 4.45 | <0.001 |
| Missing = 0 (0.00%) | | | | |
| Health decisions (*n* = 159575) | | | | |
| Alone | 26163 (16.40) | 17312 (18.67) | 8851 (13.24) | <0.001 |
| Together with partner/husband | 63640 (39.88) | 39815 (42.94) | 23825 (35.64) | |
| Husband/partner alone | 69772 (43.72) | 35600 (38.39) | 34172 (51.12) | |
| Missing = 23966 (13.06%) | | | | |

*(Continued)*

**Table 2.** (Continued)

| Variable | Pooled Sample | Weighed | Not Weighed | |
|---|---|---|---|---|
| | Frequency (%) | Frequency (%) | Frequency (%) | $x^2$/ **Diff.t-test** |
| Used ANC (*n* = 125712) | | | | |
| Yes | 109568 (87.16) | 77486 (97.95) | 32082 (68.84) | <0.001 |
| No | 16144 (12.84) | 1625 (2.05) | 14519 (31.16) | |
| Missing = 57829 (31.51%) | | | | |
| Place of delivery (*n* = 183530) | | | | |
| Health facility | 112413 (61.25) | 99857 (91.71) | 12556 (16.82) | <0.001 |
| Otherwise | 71117 (38.75) | 9028 (8.29) | 62089 (83.18) | |
| Missing = 11 (0.01%) | | | | |
| Insurance (*n* = 177413) | | | | |
| Yes | 15077 (8.50) | 12689 (12.18) | 2388 (3.26) | <0.001 |
| No | 162336 (91.50) | 91489 (87.82) | 70847 (96.74) | |
| Missing = 6128 (3.34%) | | | | |
| Distance to a facility (*n* = 183535) | | | | |
| A big problem | 71538 (38.98) | 35623 (32.72) | 35915 (48.11) | <0.001 |
| Not a big problem | 111997 (61.02) | 73265 (67.28) | 38732 (51.89) | |
| Missing = 6 (0.00%) | | | | |
| Currently working (*n* = 183,525) | | | | |
| Yes | 117785 (64.18) | 71339 (65.52) | 46446 (62.22) | |
| No | 65740 (35.82) | 37540 (34.48) | 28200 (37.78) | <0.001 |
| Currently working (*n* = 183,525) | | | | |
| **Household Characteristics** | | | | |
| Wealth status (*n* = 183541) | | | | |
| Poorest | 48380 (26.36) | 18561 (17.05) | 29819 (39.95) | <0.001 |
| Poorer | 41210 (22.45) | 20871 (19.17) | 20339 (27.25) | |
| Middle | 36739 (20.02) | 22903 (21.03) | 13836 (18.53) | |
| Wealthier | 30732 (16.74) | 23032 (21.15) | 7700 (10.31) | |
| Wealthiest | 26480 (14.43) | 23525 (21.60) | 2955 (3.96) | |
| Missing = 0 (0.00%) | | | | |

## Discussion

The study examined the factors associated with weighing a child at birth using the most recent DHS data from 16 SSA countries. We found that older women were more likely to weigh their children than younger women. Studies have found that older women have a higher chance of encountering complications during pregnancy, which could have an adverse effect on their children [25,26]. Knowing this information, older women may want to weigh their children to safeguard their health.

Educated women were more likely to weigh their children than uneducated women. This finding may be because the educated are better informed about the benefits of weighing children at birth than the uneducated. For instance, women with higher levels of education may learn about the importance of weighing children at birth. Moreover, studies have documented the correlation between education and health outcomes for women and children [27–29].

Parity was negatively correlated with the odds of weighing a child. The study revealed that the likelihood of weighing a child at birth decreases as parity increases. This finding may be linked with the observation that women with more pregnancy and birthing experience can often produce healthy babies [30,31]. Therefore, women with more parity may not see the need to weigh their children.

**Table 3. Determinants of weighing children at birth in sub-Saharan Africa.**

| Variables | Unadjusted OR | Adjusted OR |
|---|---|---|
| **Child's Characteristics** | | |
| Child sex (Ref: Male) | | |
| Female | 0.97 (0.96 - 1.00)** | 1.02 (0.97 - 1.07) |
| Birth type (Ref: Single) | | |
| Multiple | 1.21 (1.03 - 1.41)** | 1.03 (0.89 - 1.20) |
| **Mother's Characteristics** | | |
| Age at birth (Ref: 15–19) | | |
| 20–24 | 1.10 (0.95 - 1.28) | 1.14 (1.01 - 1.29)** |
| 25–29 | 1.08 (0.83 - 1.40) | 1.30 (1.03 - 1.63)** |
| 30–34 | 1.13 (0.84 - 1.50) | 1.57 (1.17 - 2.09)*** |
| 35–39 | 1.11 (0.80 - 1.55) | 1.85 (1.35 - 2.54)*** |
| 40–44 | 0.94 (0.69 - 1.27) | 1.91 (1.32 - 2.76)*** |
| 45–49 | 0.68 (0.52 - 0.90)*** | 1.67 (1.14 - 2.44)*** |
| Education level (Ref: None) | | |
| Primary | 3.11 (1.69 - 5.72)*** | 1.38 (1.19 - 1.60)*** |
| Secondary | 5.19 (3.33 - 8.07)*** | 1.64 (1.41 - 1.91)*** |
| Higher | 15.16 (8.70 - 26.41)*** | 2.68 (2.33 - 3.08)*** |
| Marital Status (Ref: Otherwise) | | |
| Married | 0.64 (0.36 - 1.13) | 1.15 (0.84 - 1.57) |
| Residence (Ref: Rural) | | |
| Urban | 3.78 (2.21 - 6.47)*** | 2.29 (1.79 - 2.91)*** |
| Parity | 0.84 (0.81 - 0.87)*** | 0.94 (0.90 - 0.98)*** |
| Health decisions (Ref: Husband/partner alone) | | |
| Woman Alone | 1.97 (1.02 - 3.82)** | 1.28 (1.04 - 1.57) ** |
| Together with partner/husband | 1.64 (0.81 - 3.32) | 1.22 (1.01 - 1.46) ** |
| Used ANC (Ref: No) | | |
| Yes | 21.76 (16.25 - 29.15)*** | 4.61 (3.76 - 5.67)*** |
| Place of delivery (Ref: Otherwise) | | |
| Health facility | 54.60 (34.47 - 86.48)*** | 52.29 (38.43 - 71.16)*** |
| Insurance (Ref: No) | | |
| Yes | 4.35 (1.48 - 12.85)*** | 1.32 (1.09 - 1.61)*** |
| Distance to health facility (Ref: big problem) | | |
| Not a big problem | 1.79 (1.14 - 2.83)** | 1.09 (1.00 - 1.19)* |
| Currently working (Ref: No) | | |
| Yes | 1.19 (0.77 - 1.83) | 1.01 (0.94 - 1.09) |
| **Household Characteristics** | | |
| Wealth status (Ref: Poorest) | | |
| Poorer | 1.51 (1.24 - 1.84)*** | 1.20 (1.07 - 1.35)*** |
| Middle | 2.40 (1.67 - 3.43)*** | 1.48 (1.22 - 1.78)*** |
| Wealthier | 4.52 (2.85 - 7.16)*** | 2.05 (1.56 - 2.70)*** |
| Wealthiest | 12.77 (7.89 - 20.67)*** | 3.91 (2.96 - 5.17)*** |
| **Countries** (Ref: Rwanda) | | |
| Angola | 0.08 (0.08 - 0.08)*** | 0.43 (0.35 - 0.53)*** |
| Benin | 0.17 (0.17 - 0.17)*** | 0.21 (0.16 - 0.28)*** |
| Cameroon | 0.11 (0.11 - 0.11)*** | 0.23 (0.19 - 0.27)*** |

*(Continued)*

**Table 3.** (Continued)

| Variables | Unadjusted OR | Adjusted OR |
|---|---|---|
| Ethiopia | 0.01 (0.01 - 0.01)*** | 0.05 (0.04 - 0.06)*** |
| Ghana | 0.10 (0.10 - 0.10)*** | 0.10 (0.09 - 0.11)*** |
| Gambia | 0.49 (0.49 - 0.49)*** | 0.87 (0.70 - 1.09) |
| Guinea | 0.06 (0.06 - 0.06)*** | 0.19 (0.15 - 0.24)*** |
| Kenya | 0.10 (0.10 - 0.10)*** | 0.22 (0.19 - 0.26)*** |
| Liberia | 0.11 (0.11 - 0.11)*** | 0.08 (0.06 - 0.10)*** |
| Madagascar | 0.03 (0.03 - 0.03)*** | 0.09 (0.08 - 0.11)*** |
| Mali | 0.06 (0.06 - 0.06)*** | 0.10 (0.07 - 0.14)*** |
| Malawi | 0.53 (0.53 - 0.53)*** | 0.57 (0.46 - 0.71)*** |
| Nigeria | 0.02 (0.02 - 0.02)*** | 0.03 (0.02 - 0.04)*** |
| Senegal | 0.21 (0.21 - 0.21)*** | 0.35 (0.27 - 0.46)*** |
| Zambia | 0.29 (0.29 - 0.29)*** | 0.35 (0.28 - 0.44)*** |
| **Interactions** | | |
| Residence & Birthplace (Ref: Rural & Otherwise) | | |
| Urban*Health facility | | 0.46 (0.36 - 0.59)*** |
| ANC & Birthplace (Ref: No ANC & Otherwise) | | |
| Used ANC* Health facility | | 0.69 (0.54 - 0.87)*** |
| Residence & Education ((Ref: Rural & No education) | | |
| Urban*Primary | | 1.13 (0.93 - 1.38) |
| Urban*Secondary | | 1.37 (1.11 - 1.70)*** |
| Urban*Higher | | 1.44 (1.07 - 1.94)** |
| Constant | | 0.13 (0.10 - 0.17)*** |
| Log pseudolikelihood | | −2.821e + 10 |
| Pseudo $R^2$ | | 0.59 |
| Observations | | 102,709 |

**Note:** OR = odds ratio; Ref = Reference group. Sampling weights and clustering (at country level) accounted for in the regression analysis. 95% confidence interval (CI) in parentheses.

***p < 0.01,

**p < 0.05,

*p < 0.10.

The results from the study further revealed that women covered by health insurance are more likely to weigh their children at birth. Health insurance may cover the cost of healthcare services, which include weighing a child, hence encouraging women, especially those with low incomes, to weigh their children. Moreover, insurance increases access to healthcare services [32–34]; hence, women with insurance may be more likely to use healthcare services such as weighing their children at birth.

The study further revealed a positive association between wealth status and weighing children at birth. The likelihood of weighing a child at birth increases with household wealth. One possible explanation behind this finding could be the cost of transportation to healthcare facilities. The poor may be unable to afford the cost of transporting their babies to healthcare facilities to have their weight checked, unlike the wealthy, who can afford it. Another possible explanation may be that low-income families prioritize meeting their fundamental daily necessities, resulting in fewer resources for healthcare (such as weighing their babies) compared to affluent households. This finding (the positive correlation between wealth and weighing of children at birth) suggests implementing anti-poverty measures to encourage weighing children at birth.

The use of ANC was also positively correlated with the weighing of children at birth. This finding may be linked to the components of ANC, which include health education and promotion, prevention and management of pregnancy-related or concurrent diseases, and risk detection [35]. Thus, women who use ANC are educated or informed about the importance of weighing children at birth, perhaps explaining why they have a higher chance of weighing their children.

Place of residence also emerged as a significant determinant of weighing children at birth. The study found that rural residents are less likely to weigh their children. Individuals who live in urban areas have better access to a range of healthcare facilities and healthcare personnel, which makes it easier for them to access healthcare services [36], including the weighing of their children.

Women who gave birth at healthcare facilities were more likely to weigh their children at birth. There are two possible explanations for this finding. The first relates to the availability of resources at the healthcare facilities, such as the tools/equipment used to weigh children and the personnel who can weigh the children. The second could be linked to the fact that women who deliver at healthcare facilities are more likely to receive postnatal care, which also includes weighing children [37,38].

One of the strengths of this study is that it is the first to investigate the factors that influence the decision to weigh a child at birth. This is a significant contribution to the literature, given that there has been no published literature on this topic. Another strength of this study is that it used nationally representative data from 16 sub-Saharan African countries to examine the factors influencing the decision to weigh a child at birth. Therefore, lessons from this study can be relevant to other countries in sub-Saharan Africa (due to the similarity in the healthcare systems and population characteristics among sub-Saharan African countries).

One limitation of this study is that potential supply-side predictors such as the healthcare facility, personnel, and weighing equipment availability were not accounted for due to data limitations. Future studies can explore the extent to which these supply-side variables influence the decision to weigh a newborn. Another limitation is that the study only establishes association, not causality. Nonetheless, the study offers a foundation for more rigorous analysis in future studies.

## Conclusion

The study investigated the factors associated with weighing a child at birth using nationally representative data from 16 SSA countries. The overall prevalence of weighing children at birth was 59%. This ranged from 27.48% in Ethiopia to 95.12% in Rwanda. Results from the study's multivariable regression showed that old age, having an education, being married, giving birth in a healthcare facility, having insurance, using ANC, living in urban areas, having autonomy in making healthcare decisions and having a higher wealth status were positively correlated (significantly) with weighing children at birth. However, higher parity was negatively correlated (significantly) with weighing children at birth.

The study's findings have some important policy implications. The finding that health facility delivery and using ANC positively correlate with weighing a child at birth suggests implementing policies/programs to encourage ANC use and facility delivery. One such policy could be creating awareness programs to educate women of childbearing age about the importance of using ANC and facility delivery. This can be done through local media, social media, and other media platforms. Furthermore, the study also provides support for women empowerment programmes. The statistical significance of autonomy in healthcare decisions and formal education suggests the need to institute policies promoting girl child education, such as scholarships for brilliant but needy girls. Moreover, the literature highlights the positive correlation between education and the autonomy of women in decision-making [39–41]. Special considerations could be given to the poor and rural residents in adopting these interventions.

## Supporting information

**S1 Appendix.  Data used for the Study.**
(ZIP)

**S2 Appendix. Test for Multicollinearity.**
(DOCX)

## Author contributions

**Conceptualization:** Alex Bawuah, Edward Nketiah-Amponsah.

**Data curation:** Alex Bawuah, Samuel Ampaw.

**Formal analysis:** Alex Bawuah, Samuel Ampaw, Edward Nketiah-Amponsah.

**Investigation:** Alex Bawuah, Edward Nketiah-Amponsah.

**Methodology:** Alex Bawuah, Samuel Ampaw, Edward Nketiah-Amponsah.

**Project administration:** Alex Bawuah.

**Resources:** Alex Bawuah, Samuel Ampaw.

**Supervision:** Alex Bawuah.

**Validation:** Alex Bawuah, Edward Nketiah-Amponsah.

**Visualization:** Alex Bawuah, Samuel Ampaw, Edward Nketiah-Amponsah.

**Writing – original draft:** Alex Bawuah.

**Writing – review & editing:** Alex Bawuah, Samuel Ampaw, Edward Nketiah-Amponsah.

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
