## [Decision Letter · Decision Letter 0]

Dear Dr. Nketiah-Amponsah,

Thank you for submitting your manuscript to PLOS ONE. After careful consideration, we feel that it has merit but does not fully meet PLOS ONE’s publication criteria as it currently stands. Therefore, we invite you to submit a revised version of the manuscript that addresses the points raised during the review process.

We look forward to receiving your revised manuscript.

Kind regards,

George Kuryan

Academic Editor

PLOS ONE

Journal Requirements:

Reviewers' comments:

Reviewer's Responses to Questions

**Comments to the Author**

1. Is the manuscript technically sound, and do the data support the conclusions?

Reviewer #1: Yes

2. Has the statistical analysis been performed appropriately and rigorously?

Reviewer #1: Yes

3. Have the authors made all data underlying the findings in their manuscript fully available?

Reviewer #1: No

4. Is the manuscript presented in an intelligible fashion and written in standard English?

Reviewer #1: Yes

**Reviewer #1:**  I appreciate the authors highlighting health disparities by emphasizing the importance of weighing newborn babies, and how it will help health organizations and policy administrations address neonatal complications and mortalities.

Introduction: No changes required

Methods:

• The study conducted both unadjusted and adjusted analyses. In the adjusted analysis, did the authors explore whether there were any interaction effects?

• Additionally, did the authors identify any high correlations between independent variables? For example, are women who are wealthy more likely to live in urban areas and also be well-educated?

Results and Discussion:

• Page 5, line 21 to 27: I see which African country has the highest and lowest percentage of children weighted at birth. However, I don’t see authors mentioning why that is the case.

• This also raises the question of why the authors did not include the country as an independent variable in the adjusted analysis. Despite the similarities within sub-Saharan countries, health policy and care delivery might differ among these countries, potentially affecting the final response variable, i.e., recording infant weight after birth.

• Table 1 – consider removing the column ‘% of children not weighted at birth’ as it seems redundant.

• The numbers in Table 2 don’t add up to 100%. There is a small margin of error for some categories. For example, when adding the percentages for 'weighed' and 'not weighed' for both male and female genders, it does not exactly equal the pooled sample total. The same issue arises for birth type. I encourage the authors to double-check the numbers."

Conclusion: no changes required.

**Do you want your identity to be public for this peer review?** For information about this choice, including consent withdrawal, please see our Privacy Policy

Reviewer #1: No

---

## [Author Response · Author response to Decision Letter 1]

14 May 2024

Dear Editor,

Thank you sincerely for the positive feedback and the meticulous review and constructive comments on our manuscript. The comments have been invaluable in improving the quality of our manuscript.

In response to the reviewer comments, we have diligently implemented the necessary corrections throughout the manuscript. It is important to note that these revisions may have resulted in changes to line and page numbers. Our commitment to enhancing the clarity and precision of our submission has been paramount in making these adjustments.

Please find below our responses to the reviewers’ comments; the comments are italicized while the responses are highlighted in read. Moreover, the ensuing corrections or changes in the manuscript are tracked for easy recognition.

Reviewer #1:

Comment

I appreciate the authors highlighting health disparities by emphasizing the importance of weighing newborn babies, and how it will help health organizations and policy administrations address neonatal complications and mortalities.

Response

No changes required. Many thanks for the compliment.

Comment

Methods:

The study conducted both unadjusted and adjusted analyses. In the adjusted analysis, did the authors explore whether there were any interaction effects?

Response

Many thanks for this insightful suggestion. Accordingly, we have included interactions in the adjusted analysis. We interacted place of residence and birthplace, using ANC and birthplace and education and place of residence (see the revised estimates in Table 3 for the results on pages 9 and 10).

Comments

Additionally, did the authors identify any high correlations between independent variables? For example, are women who are wealthy more likely to live in urban areas and also be well-educated?

Response

Thank you for drawing our attention to the need to provide evidence on the extent of collinearity between the independent variables. We performed a diagnostic test for collinearity between the independent variables by looking at the variance inflation factor (VIF) and tolerance indices (TI) of the independent variables (see page 5 lines 20 to 26).

Comment

Results and Discussion:

• Page 5, line 21 to 27: I see which African country has the highest and lowest percentage of children weighted at birth. However, I don’t see authors mentioning why that is the case.

• This also raises the question of why the authors did not include the country as an independent variable in the adjusted analysis. Despite the similarities within sub-Saharan countries, health policy and care delivery might differ among these countries, potentially affecting the final response variable, i.e., recording infant weight after birth.

Response

We have addressed this concern by including the country as an independent variable in the adjusted analysis. See table 3 for the results on pages 9 and 10 and page 12 line 20 to 26 for possible explanation for the differences in the percentages of children weighed at birth across the sampled countries. For instance, in 2006, Rwanda launched a national Facility-Based Childbirth Policy (FBCP), which provides free antenatal care and facility-based delivery services. The policy led to an increase in intuitional delivery and other maternal healthcare services. Perhaps, the reason why children in Rwanda are more likely to be weighed compared to children from other countries.

Comment

Table 1 – consider removing the column ‘% of children not weighted at birth’ as it seems redundant.

Response

Many thanks. This has been duly deleted.

Comment

The numbers in Table 2 don’t add up to 100%. There is a small margin of error for some categories. For example, when adding the percentages for 'weighed' and 'not weighed' for both male and female genders, it does not exactly equal the pooled sample total. The same issue arises for birth type. I encourage the authors to double-check the numbers."

Response

Thanks for drawing our attention to this anomaly. We have made the requisite changes.

We hope our revision will meet your expectations.

Yours faithfully

Edward Nketiah-Amponsah (On behalf of co-authors)

---

## [Decision Letter · Decision Letter 1]

Dear Dr. Nketiah-Amponsah,

Thank you for submitting your manuscript to PLOS ONE. After careful consideration, we feel that it has merit but does not fully meet PLOS ONE’s publication criteria as it currently stands. Therefore, we invite you to submit a revised version of the manuscript that addresses the points raised during the review process.

We look forward to receiving your revised manuscript.

Kind regards,

Gbenga Kayode

Academic Editor

PLOS ONE

Additional Editor Comments:

Although both reviewers recommended the manuscript for acceptance, I would like the authors to address the following comments before we consider publication.

1) The authors should explain why the year of data collection was not included in the analysis.

2) Given the hierarchical nature of the data, the authors should clarify why a hierarchical or multilevel model was not used to analyze the data.

3) The authors should clarify why health indicators from previous births were not considered.

4) For each variable in Table 2, the authors should present the percentages of missing data.

5) The authors should clarify why they did not consider the weight of the data in their analysis.

6) The authors should clarify why significant factors like employment status and birth order were excluded from the analysis.

7) The authors should clarify how they handled missing data.

Reviewers' comments:

Reviewer's Responses to Questions

**Comments to the Author**

Reviewer #1: All comments have been addressed

Reviewer #2: All comments have been addressed

2. Is the manuscript technically sound, and do the data support the conclusions?

Reviewer #1: Yes

Reviewer #2: Yes

3. Has the statistical analysis been performed appropriately and rigorously?

Reviewer #1: Yes

Reviewer #2: Yes

4. Have the authors made all data underlying the findings in their manuscript fully available?

Reviewer #1: Yes

Reviewer #2: Yes

5. Is the manuscript presented in an intelligible fashion and written in standard English?

Reviewer #1: Yes

Reviewer #2: Yes

Reviewer #1: Page 12 line 21, there is a typo, change it to “countries”. Otherwise, the current manuscript draft reads well, and I acknowledge that all my comments have been addressed. I wish authors good luck for their future research works.

Thanks!

Reviewer #2: The manuscript have addressed all the reviewer's comments. The write up is concise and comprehensive . However, i will recommend the word"lilkelier be replaced with "most likely', otherwise, the manuscript was well written. The introduction, objectives, methodology, results and discussion were all well captured.

**Do you want your identity to be public for this peer review?** For information about this choice, including consent withdrawal, please see our Privacy Policy

Reviewer #1: No

Reviewer #2: No

---

## [Author Response · Author response to Decision Letter 2]

22 Feb 2025

Response to Editor's Comments

We wish to thank you and the reviewers sincerely for your constructive comments on our manuscript titled ‘Factors associated with weighing a child at birth: evidence from 16 sub-Saharan African countries’ submitted to PLOS ONE for publication consideration. The comments have been invaluable in improving the quality of our manuscript.

In response to the reviewer comments, we have diligently implemented the necessary corrections throughout the manuscript. We have addressed the comments point-by-point as far as feasible. Please find below our responses to the reviewers’ comments; the comments are boldened while the responses are italicized. Moreover, the ensuing corrections or changes in the manuscript are identified by page numbers for easy recognition whenever applicable.

1. The authors should explain why the year of data collection was not included in the analysis.

Response: The model initially included the year variable to account for potential temporal effects. However, it (year) was perfectly collinear with the country variable. Given this collinearity, the year variable was omitted from the final model. We can still show the results with the year variable if deemed necessary.

2. Given the hierarchical nature of the data, the authors should clarify why a hierarchical or multilevel model was not used to analyse the data.

Response: We acknowledge the hierarchical nature of the data, where individuals are nested within countries. To address this, we accounted for clustering and country-level heterogeneity in the following ways;

i. Clustering at the Country Level: We used clustered standard errors at the country level (vce(cluster Country)) to adjust for within-country correlation. This ensures that our standard errors are robust to potential intra-country dependencies.

ii. Country-Fixed Effects: We included country-fixed effects (i.Country) in our regression model to control for unobserved country-level characteristics that might influence the outcome variable. This approach allows us to isolate within-country variation while accounting for systematic differences across countries.

3. The authors should clarify why health indicators from previous births were not considered.

Response: Health indicators from previous births are only available for women who have had more than one birth. Including these variables would exclude women with only one birth from the analysis. Since our study is not limited to women with multiple births, we chose not to incorporate these indicators to maintain a broader and more representative and consistent sample.

4. For each variable in Table 2, the authors should present the percentages of missing data.

Response: This has been duly provided in the revised version of the manuscript. See Table 2 (pages 7 and 8)

5. The authors should clarify why they did not consider the weight of the data in their analysis.

Response: Our analysis accounted for survey sampling weights by specifying the probability weights (pweight=v005) in our regression model. See page 5, line 24.

6. The authors should clarify why significant factors like employment status and birth order were excluded from the analysis.

Response: We have included employment status (currently working variable) in the revised version of the manuscript. The model did not include birth order due to collinearity with the parity variable. We are happy to include it regardless, if deemed necessary.

7. The authors should clarify how they handled missing data.

Response: We handled missing data using listwise deletion. Thus, observations with missing values in any of the variables included in the regression model were excluded from the analysis. This approach ensures consistency across all model estimates. See page 5, lines 24 to 27.

We hope our revisions will meet your kind expectations.

Yours faithfully,

Prof. Edward Nketiah-Amponsah, PhD (on behalf of all co-authors)

---

## [Decision Letter · Decision Letter 2]

Dear Dr. Nketiah-Amponsah,

Thank you for submitting your manuscript to PLOS ONE. After careful consideration, we feel that it has merit but does not fully meet PLOS ONE’s publication criteria as it currently stands. Therefore, we invite you to submit a revised version of the manuscript that addresses the points raised during the review process.

We look forward to receiving your revised manuscript.

Kind regards,

Alfredo Luis Fort, M.D., M.Sc., Ph.D.

Academic Editor

PLOS ONE

Journal Requirements:

Reviewers' comments:

Reviewer's Responses to Questions

**Comments to the Author**

Reviewer #2: All comments have been addressed

2. Is the manuscript technically sound, and do the data support the conclusions?

Reviewer #2: Yes

3. Has the statistical analysis been performed appropriately and rigorously?

Reviewer #2: Yes

4. Have the authors made all data underlying the findings in their manuscript fully available?

Reviewer #2: Yes

5. Is the manuscript presented in an intelligible fashion and written in standard English?

Reviewer #2: Yes

Reviewer #2: I have reviewed the revised manuscript and confirm that the author has satisfactorily addressed all the concerns and suggestions raised during the review process. The revisions have strengthened the clarity, scientific rigor, and relevance of the study. The responses to comments were clear, and the updated manuscript reflects the necessary improvements in content and structure.

I find no further issues requiring revision and recommend the manuscript for acceptance in its current form.

**Do you want your identity to be public for this peer review?** For information about this choice, including consent withdrawal, please see our Privacy Policy

Reviewer #2: **Yes: ** Fatima Abubakar Ishaq

---

## [Author Response · Author response to Decision Letter 3]

12 Jun 2025

We sincerely thank you for your constructive editorial comments on our manuscript titled ‘Factors associated with weighing a child at birth: evidence from 16 sub-Saharan African countries’ submitted to PLOS ONE for publication consideration. The comments have been invaluable in improving the quality of our manuscript.

In particular, we requested a colleague who is more proficient in the English language to proofread the manuscript. We have also diligently addressed all the minor but essential errors, such as omitting the percent (%) sign in some figures. Besides, we have accepted your suggested corrections in tracked changes across the manuscript. For ease of reference, we have provided the clean and tracked changes versions of the manuscript for your review.

We hope our revisions will meet your kind expectations.

---

## [Editor Report · Decision Letter 3]

Factors associated with weighing a child at birth: evidence from 16 sub-Saharan African countries

PONE-D-24-00704R3

Dear Dr. Nketiah-Amponsah,

We’re pleased to inform you that your manuscript has been judged scientifically suitable for publication and will be formally accepted for publication once it meets all outstanding technical requirements.

Kind regards,

Alfredo Luis Fort, M.D., M.Sc., Ph.D.

Academic Editor

PLOS ONE

Additional Editor Comments (optional): I have added a copy of the latest version with a few suggested additions and edits to make the manuscript more understandable to general readers. Please see them in the attached file. 

---

## [Editor Report · Acceptance letter]

PONE-D-24-00704R3

PLOS ONE

Dear Dr. Nketiah-Amponsah,

I'm pleased to inform you that your manuscript has been deemed suitable for publication in PLOS ONE. Congratulations! Your manuscript is now being handed over to our production team.

Kind regards,

on behalf of

Dr. Alfredo Luis Fort

Academic Editor

PLOS ONE